# Remdesivir in Very Old Patients (≥80 Years) Hospitalized with COVID-19: Real World Data from the SEMI-COVID-19 Registry

**DOI:** 10.3390/jcm11133769

**Published:** 2022-06-29

**Authors:** Jose-Manuel Ramos-Rincon, María-Dolores López-Carmona, Lidia Cobos-Palacios, Almudena López-Sampalo, Manuel Rubio-Rivas, María-Dolores Martín-Escalante, Santiago de-Cossio-Tejido, María-Luisa Taboada-Martínez, Antonio Muiño-Miguez, Maria Areses-Manrique, Carmen Martinez-Cilleros, Carlota Tuñón-de-Almeida, Lucy Abella-Vázquez, Angel-Luís Martínez-Gonzalez, Luis-Felipe Díez-García, Carlos-Jorge Ripper, Victor Asensi, Angeles Martinez-Pascual, Pablo Guisado-Vasco, Carlos Lumbreras-Bermejo, Ricardo Gómez-Huelgas

**Affiliations:** 1Department of Clinical Medicine, Miguel Hernandez University of Elche, Ctra N332 s/n, 03550 Alicante, Spain; 2Department of Internal Medicine, Instituto de Investigacion Biomedica de Málaga (IBIMA), Regional University Hospital of Málaga, 29010 Málaga, Spain; mdlcorreo@gmail.com (M.-D.L.-C.); cobospalacios@gmail.com (L.C.-P.); almu_540@hotmail.com (A.L.-S.); ricardogomezhuelgas@hotmail.com (R.G.-H.); 3Department of Medicine, University of Málaga, 29010 Málaga, Spain; 4Internal Medicine Department, Bellvitge University Hospital, L’Hospitalet de Llobregat, 08907 Barcelona, Spain; mrubio@bellvitgehospital.cat; 5Internal Medicine Department, Costa del Sol Hospital, 29603 Marbella, Spain; mmartinescalante@gmail.com; 6Internal Medicine Department, Doce de Octubre University Hospital, 28041 Madrid, Spain; santiagodecossio@gmail.com (S.d.-C.-T.); clumbrerasb@gmail.com (C.L.-B.); 7Internal Medicine Department, Cabueñes University Hospital, 33394 Gijón, Spain; mltmartinez72@gmail.com; 8Internal Medicine Department, Gregorio Marañón General University Hospital, 28007 Madrid, Spain; antonio.muino@madrid.org; 9Internal Medicine Department, Santa Marina Hospital, 48004 Bilbo, Spain; maria.aresesmanrique@osakidetza.eus; 10Internal Medicine Department, HLA Moncloa University Hospital, 28008 Madrid, Spain; cmcilleros@hotmail.com; 11Internal Medicine Department, Zamora Hospital Complex, 49022 Zamora, Spain; carlottadealmeida@gmail.com; 12Internal Medicine Department, Nuestra Señora Candelaria University Hospital, 38010 Santa Cruz de Tenerife, Spain; abellavazquez@gmail.com; 13Internal Medicine Department, Leon University Hospital Complex, 24071 León, Spain; garufa125@gmail.com; 14Internal Medicine Department, Torrecárdenas University Hospital, 04009 Almería, Spain; lfdiez@telefonica.net; 15Internal Medicine Department, Insular University Hospital Complex, 35016 Las Palmas de Gran Canaria, Spain; cjripper@hotmail.com; 16Internal Medicine Department, Central Asturias, University Hospital, 33011 Oviedo, Spain; vasensia@gmail.com; 17Internal Medicine Department, Francesc de Borja Hospital, 46702 Gandia, Spain; mangelesmarpas@gmail.com; 18Internal Medicine Department, Quironsalud A Coruña Hospital, 15009 A Coruña, Spain; pablogvasco@gmail.com; 19CIBER Fisiopatologia de la Obesidad y la Nutricion, Carlos III Health Institute, 28029 Madrid, Spain

**Keywords:** COVID-19, SARS-CoV-2, age ≥ 80, remdesivir, mortality, Spain

## Abstract

(1) Background: Large cohort studies of patients with COVID-19 treated with remdesivir have reported improved clinical outcomes, but data on older patients are scarce. Objective: This work aims to assess the potential benefit of remdesivir in unvaccinated very old patients hospitalized with COVID-19; (2) Methods: This is a retrospective analysis of patients ≥ 80 years hospitalized in Spain between 15 July and 31 December 2020 (SEMI-COVID-19 Registry). Differences in 30-day all-cause mortality were adjusted using a multivariable regression analysis. (3) Results: Of the 4331 patients admitted, 1312 (30.3%) were ≥80 years. Very old patients treated with remdesivir (n: 140, 10.7%) had a lower mortality rate than those not treated with remdesivir (OR (95% CI): 0.45 (0.29–0.69)). After multivariable adjustment by age, sex, and variables associated with lower mortality (place of COVID-19 acquisition; degree of dependence; comorbidities; dementia; duration of symptoms; admission qSOFA; chest X-ray; D-dimer; and treatment with corticosteroids, tocilizumab, beta-lactams, macrolides, and high-flow nasal canula oxygen), the use of remdesivir remained associated with a lower 30-day all-cause mortality rate (adjusted OR (95% CI): 0.40 (0.22–0.61) (*p* < 0.001)). (4) Conclusions: Remdesivir may reduce mortality in very old patients hospitalized with COVID-19.

## 1. Introduction

Remdesivir (GS-5734), a nucleotide analog prodrug that inhibits the SARS-CoV-2 RNA-dependent RNA polymerase, has been studied in patents with COVID-19 in multiple clinical trials and cohort studies [1]. Initial randomized placebo-controlled trials showed a faster time to recovery with remdesivir, but no survival benefit was demonstrated [2,3]. Two large randomized trials, the Solidarity and Discovery trials, did not show a clinical benefit with remdesivir in patients hospitalized for COVID-19 [4,5]. Subsequent open-label randomized trials and cohort studies have yielded mixed results. Some large cohort studies have shown an improvement in clinical outcomes with remdesivir [6,7,8,9]. More recently, among outpatients at high risk for COVID-19 progression, a three-day course of remdesivir resulted in an 87% lower risk of death or hospitalization than a placebo, with an acceptable safety profile [10].

Remdesivir was approved by the United States of America’s Food and Drug Administration and the European Medicines Agency for the treatment of patients hospitalized with COVID-19 with an oxygen saturation ≤ 94% on room air or those who require supplemental oxygen. It is well-established that very old patients with COVID-19 are at high risk of mortality [11,12,13]. However, the effectiveness and safety of remdesivir among elderly patients with moderate–severe COVID-19 in clinical practice remains unclear [14]. Therefore, it is critical to improve knowledge about the potential role of remdesivir in older patients with COVID-19.

This work aims to assess the potential benefit of remdesivir on clinical outcomes among unvaccinated very old patients (≥80 years) hospitalized with COVID-19 included in a large nationwide Spanish registry.

## 2. Materials and Methods

### 2.1. Study Design and Population

This is a retrospective cohort study of patients ≥ 80 years hospitalized with COVID-19 in Spain from 1 July to 31 December 2020. The data source was the SEMI-COVID-19 Registry. In Spain, COVID-19 vaccination started in January 2021, so all patients included in this study were unvaccinated.

### 2.2. Definition of Variables

All patient data were drawn from the Spanish Society of Internal Medicine’s SEMI-COVID-19 Registry, which had the participation of 150 Spanish hospitals. The SEMI-COVID-19 Registry prospectively collects data on the first admission of patients ≥ 18 years with COVID-19 confirmed microbiologically via a reverse transcription polymerase chain reaction (RT-PCR) or antigen test. More detailed information on the justification, aims, methods, and preliminary outcomes of the SEMI-COVID-19 Registry are available in previously published works [15].

The Barthel Index was used to evaluate the degree of dependence. Comorbidities were evaluated by means of the age-adjusted Charlson Comorbidity Index (CCI) [16]. Patients were determined to have diabetes mellitus, dyslipidemia, or hypertension if there was a prior diagnosis on their electronic medical record (EMR) or if they had pharmacological treatment for these diseases. Atherosclerotic cardiovascular disease was defined as a history of coronary heart disease (acute coronary syndrome, angina, coronary revascularization, or myocardial infarction), cerebrovascular disease (transient ischemic attack, stroke), or peripheral arterial disease (revascularization, intermittent claudication, abdominal aortic aneurysm, or lower limb amputation). Chronic pulmonary disease was considered present if the patient had been diagnosed with asthma and/or chronic obstructive pulmonary disease. Malignancy included solid tumors (excluding non-melanoma skin cancer) and/or hematologic neoplasia. Data on all baseline comorbidities were collected from EMR in the hospitals. The laboratory data (metabolic panel, complete blood count, blood gases, coagulation) and diagnostic imaging tests were taken upon admission.

According to the Spanish Agency of Medicines and Medical Devices, treatment with remdesivir is indicated for patients hospitalized with COVID-19 who meet the following criteria: (1) age > 12 years and weight > 40 kg; (2) need for supplemental low-flow oxygen; (3) ≤7 days from symptoms onset to the prescribing of remdesivir; and (4) meet at least two of the following three criteria: respiratory rate ≥ 24 bpm, oxygen saturation ≤ 94% on room air, or PaO_2_ /FiO_2_ < 300 mmHg. As per AEMPS guidelines, the regimen for intravenous remdesivir is 200 mg on day 1 and 100 mg on days 2 through 5. The treatments used during admission were antimicrobial therapy (beta-lactams, macrolides, or quinolones), immunomodulatory therapy (systemic corticosteroids, tocilizumab), or anticoagulant therapy (oral anticoagulants or low-molecular-weight heparin). Hydroxychloroquine, chloroquine, and lopinavir/ritonavir were not used during the study period.

The study’s primary endpoint was 30-day all-cause mortality. Other endpoints analyzed were admission to the intensive care unit (ICU), use of invasive mechanical ventilation (IMV), length of stay (LOS), and readmission within 30 days of hospital discharge.

### 2.3. Statistical Analysis

Patients’ characteristics were analyzed using descriptive statistics. Continuous and categorical variables were shown, respectively, as medians and interquartile ranges (IQRs) and as absolute values and percentages. Differences among groups were analyzed via the Mann–Whitney U test for continuous variables and Pearson’s chi-square test for categorical variables. Statistical significance was established as *p* < 0.05.

Differences in 30-day all-cause mortality between patients treated or not treated with remdesivir were adjusted by age and sex. Variables found to be statistically significant on a bivariate analysis were included in a multivariate regression analysis using a stepwise regression with a threshold of *p* < 0.10. The values were shown as adjusted odds ratios (ORs) and 95% confidence intervals (CIs). IBM SPSS Statistics v25 (Armonk, NY, USA) was used for the statistical analyses.

### 2.4. Ethical Aspects

The Institutional Research Ethics Committee of Málaga, Spain approved this work on 27 March 2020 (Ethics Committee code: SEMI-COVID-19 27-03-20), pursuant to Spanish Agency of Medicines and Medical Products guidelines. All patients provided their informed consent. All data in this work that were collected, processed, and analyzed were anonymized and used solely for the purposes of this work. All data were protected pursuant to Regulation (EU) 2016/679 of the European Parliament and of the Council of 27 April 2016 on the protection of natural persons with regard to the processing of personal data and on the free movement of such data. The Institutional Research Ethics Committees of each participating hospital also approved this work.

## 3. Results

### 3.1. Use of Remdesivir

Of the 4331 patients admitted during the study period, 1312 (30.3%) were ≥80 years. Of them, remdesivir was used in 140 patients (10.7%). The median duration of symptoms prior to starting therapy was 5 days (IQR: 4–7). Regarding the duration of treatment with remdesivir, 18 (11.1%) patients were treated for ≤3 days, 107 (78.6%) for 4–5 days, and 11 (8.0%) for >5 days.

### 3.2. Differences in the Profile of Patients Aged ≥80 Years Treated or Not Treated with Remdesivir

The median age of patients treated with remdesivir was slightly lower (85 (IQR: 83–89) vs. 86 (IQR: 83–90) years, *p* = 0.049). Among patients treated with remdesivir, fewer cases were admitted from nursing homes (7.1% vs. 24.4%, *p* = 0.003), fewer had moderate or severe dependence (17.9% and 9.3% vs. 32.1% and 32.1%, *p* < 0.001) or dementia (8.76% vs. 32.1%, *p* < 0.001), and the mean CCI was lower (5.8 vs. 6.3, *p* = 0.001) compared to patients not treated with remdesivir. Moreover, fewer patients treated with remdesivir had a qSOFA score ≥ 2 at admission (5.0% vs. 16.0%, *p* = 0.001). The prevalence of oxygen saturation ≤ 94% or tachypnea was similar among both group of patients. Upon admission, more patients treated with remdesivir had bilateral infiltrates on a chest X-ray (66.2% vs. 53.8%, *p* = 0.02) and a lower mean D-dimer level (890 vs. 1020 ng/mL, *p* = 0.005). More patients treated with remdesivir received systemic corticosteroids (90.0% vs. 79.4%; *p* = 0.002) and tocilizumab (15% vs. 4.3%, *p* < 0.001) and fewer received beta-lactams (63.6% vs. 72.3%, *p* = 0.030) and macrolides (24.3% vs. 40.2%, *p* < 0.001). Finally, more remdesivir-treated patients received high-flow nasal cannula oxygen (15.7% vs. 4.6%, *p* < 0.001). These data are shown in Table 1.

### 3.3. Clinical Outcomes in Patients ≥ 80 Years Treated with Remdesivir

The 30-day all-cause mortality rates in patients treated and not treated with remdesivir were 20.0% and 35.7%, respectively (OR: 0.45, 95% CI: 0.29–0.69, *p* < 0.001). ICU admission was rare, but higher in patients treated with remdesivir (3.6% vs. 1.3%, *p* = 0.036). The median LOS was longer in those treated with remdesivir (13 vs. 9 days, *p* < 0.001). There were no differences in 30-day readmissions between patients treated and not treated with remdesivir. These data are shown in Table 2.

A lower mortality rate was observed among remdesivir-treated patients. This lower rate persisted after adjusting for age and sex in a multivariate analysis and after adjusting for all variables found to be significant in the bivariate analysis (place of COVID-19 acquisition, dependence, baseline CCI, dementia, duration of symptoms, qSOFA, chest X-ray, D-dimer, systemic corticosteroids, tocilizumab, beta-lactams, macrolides, and high-flow nasal canula oxygen) (adjusted OR: 0.40, 95% CI: 0.22–0.61, *p* < 0.0001). These data are shown in Table 3.

## 4. Discussion

This work assesses the efficacy of remdesivir in a real-life cohort of very old patients hospitalized with COVID-19 in Spain prior to the start of the vaccination campaign. We found that patients ≥ 80 years who received remdesivir showed a 15.7% lower 30-day all-cause mortality rate and a 60% reduction in the adjusted risk of mortality compared to non-treated patients.

The clinical trials and observational studies that evaluate the efficacy of remdesivir in COVID-19 have shown conflicting results regarding reductions in in-hospital mortality, LOS, and ICU admission. Several observational studies in the general population do not support the use of remdesivir for improving clinical recovery and decreasing mortality due to SARS-CoV-2 infection [2,7]. In other studies, treatment with remdesivir was associated with a lower ICU admission rate and shorter LOS [9,17]. In another observational study, remdesivir treatment did not increase survival and was associated with a longer LOS [18]. Finally, in a small, single-center study conducted in Spain, the use of remdesivir in hospitalized patients with COVID-19 was associated with a lower mortality rate [19].

A recent metanalysis by Ansema et al. [20] that included five randomized clinical trials concluded that remdesivir probably has little or no effect on 28-day all-cause mortality in hospitalized adults with SARS-CoV-2 infection. However, there were not enough data available to examine the effect of remdesivir on mortality in subgroups based on the extent of baseline respiratory support. Another open-label randomized clinical trial showed that remdesivir led to a modest but significant decline in mortality compared to standard of care [21]. A recent clinical trial analyzing unhospitalized patients at high risk for COVID-19 progression found that a three-day remdesivir course led to a 87% lower risk of death or hospitalization than a placebo, with an acceptable safety profile [10].

Interestingly, there are no studies on remdesivir that have specifically focused on very old patients, even though this subgroup has had the highest rate of mortality during the pandemic [14]. Our study suggests that remdesivir can reduce mortality in very old patients hospitalized with COVID-19. Although patients treated with remdesivir in our series had fewer comorbidities (less functional dependence; fewer days of symptoms; and more use of systemic corticosteroids, tocilizumab, and high-flow nasal cannula oxygen), this benefit in mortality persisted after adjusting for all these confounding variables.

Certain risk factors, such as dependence and the presence of dementia, are associated with higher mortality in very old patients. [11,22]. In our study, few patients with moderate or severe dependence and dementia received remdesivir and they were likely a group of people who should have received the drug. Parkinson’s disease and parkinsonism are risk factors for worse outcomes in patients with COVID-19, as has been shown in the meta-analysis by Putri C. et al. [23], and these patients would probably have benefited from treatment. However, data on Parkinson’s disease and parkinsonism were not collected in this study.

After the publication of the outcomes of the Recovery trial, which showed a significant reduction in mortality, dexamethasone became the standard of care for hospitalized patients with COVID-19 pneumonia who require oxygen [24]. Moreover, an observational retrospective study showed that remdesivir plus corticosteroid administration did not reduce the time to death compared to remdesivir administered alone. In our study, most patients were treated with corticosteroids and the use of corticosteroids was associated with a lower mortality rate, but the potential benefit of remdesivir persisted even after adjusting for the use of corticosteroids.

In contrast, a randomized clinical trial showed that baricitinib plus remdesivir was superior to remdesivir monotherapy in reducing time to recovery and accelerating improvement in clinical status among COVID-19 patients, particularly among patients receiving high-flow oxygen or NIV [22]. However, in our study, only four patients were treated with baricitinib, so it was not possible to draw conclusions.

It is unclear whether the good outcomes observed in elderly patients treated with remdesivir could be due to immunosenescence in this population [25]. If so, the earlier use of antiviral therapy in these patients could be another explanation of the better response to antivirals, in light of the fact that they accessed medical care earlier [26], when the efficacy of treatment is higher.

This work shows that remdesivir treatment in elderly patients was associated with lower mortality and that antiviral treatment in the first few days of illness reduced mortality in unvaccinated patients. Therefore, it can be expected that in vaccinated patients, which are the majority of those currently admitted to the hospital for COVID-19, treatment with remdesivir may improve the course of the disease in elderly patients. The main strength of our study is the large sample size of very old patients with COVID-19, an age subgroup that has not been well studied. Another strength is that the indications for the use of remdesivir were very uniform according to strict guidelines from the Spanish Ministry of Health. Finally, our database (SEMI-COVID-19 Registry) is a high-quality registry endorsed by multiple publications.

This investigation has several limitations. First, as with any observational study, we cannot entirely rule out the effect of treatment selection bias or residual or unobserved confounding factors, despite having performed a multivariate regression model. Second, our study was limited to unvaccinated patients admitted during the first two waves of the pandemic, and as such we cannot extrapolate our results to the current epidemiological scenario in Spain, in which most older adults are vaccinated. Finally, we did not analyze the potential adverse effects of the use of remdesivir in very old patients. Nevertheless, in a study by Kanai et al. [14], remdesivir was discontinued due to adverse events in less than 4% of older patients.

## 5. Conclusions

In conclusion, our study suggests that remdesivir may reduce mortality in very old patients hospitalized with COVID-19. Though a range of therapeutic approaches, including multiple immunotherapy agents, novel antivirals, and combination treatments, are necessary, our findings highlight the potential important role of antivirals in very old patients. More specific research on therapeutic strategies for COVID-19 in very old patients is needed.

## Figures and Tables

**Table 1 jcm-11-03769-t001:** Baseline characteristics of very old patients (≥80 years) hospitalized with COVID-19 treated or not treated with remdesivir.

	Remdesivir (n = 140)	No Remdesivir (n = 1172)	*p* Value
**Sociodemographic variables**			
Age (years), median (IQR)	85 (83–89)	86 (83–90)	**0.049**
Sex (male), n (%)	72 (51.4)	560 (47.8)	0.414
**Acquisition**, n (%)			**<0.001**
Community	113 (80.7)	813 (69.4)	
Nosocomial	17 (12.1)	72 (6.1)	
Nursing home	10 (7.1)	286 (24.4)	
**Degree of dependence**, n (%)			**<0.001**
Independent or mild	102 (72.9)	484 (41.5)	
Moderate	25 (17.9)	374 (32.1)	
Severe	13 (9.3)	308 (26.4)	
**Comorbidities**			
Baseline CCI, median (IQR)	6 (5–7)	6 (5–7)	**0.001**
Baseline CCI ≥ 6, n (%)	63 (45.7)	435 (37.6)	0.066
Hypertension	109 (77.9)	920 (78.5)	0.862
Non-atherosclerotic cardiovascular disease ^a^	47 (33.6)	392 (33.5)	0.993
Atherosclerotic cardiovascular disease ^b^	50 (35.7)	361 (30.9)	0.244
Dementia	12 (8.6)	376 (32.1)	<0.001
Diabetes mellitus	42 (30.0)	372 (31.8)	0.671
Chronic pulmonary disease ^c^	28 (20.0)	240 (20.5)	0.887
Obesity ^f^	32 (22.9)	178 (17.2)	0.102
Malignancy ^d^	17 (12.1)	163 (14.0)	0.556
Moderate-to-severe kidney disease ^e^	13 (9.3)	149 (12.7)	0.244
**Symptoms and physical examination**			
Duration of symptoms in days, median (IQR)	4 (1–5)	4 (1–7)	**0.012**
Oxygen saturation ≤ 94%, n (%)	66 (47.8)	541 (47.0)	0.862
Hypotension, n (%)	4 (2.9)	83 (7.2)	0.056
Tachypnea, n (%)	58 (41.7)	416 (35.7)	0.163
Tachycardias, n (%)	20 (14.3)	205 (17.7)	0.313
qSOFA index ≥ 2, n (%)	7 (5.0)	188 (16.0)	**0.001**
**Chest X-ray findings**, n (%)			**0.020**
Normal	26 (18.7)	280 (24.0)	
Unilateral infiltrates	21 (15.1)	259 (22.2)	
Bilateral infiltrates	92 (66.2)	628 (53.8)	
**Laboratory findings**, n (%)			
PO_2_/FiO_2_ ratio	287 (238–332)	287 (22–328)	0.973
Lymphocytes (×10^3^/µL)	0.85 (0.69–1.26)	0.89 (0.60–1.23)	0.473
Lactate dehydrogenase (U/L)	303 (247–416)	301 (228–406)	0.196
C-reactive protein (mg/L)	77 (21–127)	70 (28–128)	0.803
D-dimer (ng/mL)	890 (416–1499)	1020 (585–2067)	**0.005**
Serum ferritin (µg/L)	550 (243–904)	374 (180–781)	0.042
Fibrinogen (mg/L)	558 (460–693)	558 (460–693)	0.447
**Other treatment**, n (%)			
Systemic corticosteroids	126 (90.0)	926 (79.1)	**0.002**
Tocilizumab	21 (15.0)	50 (4.3)	**<0.001**
Baricitinib	1 (0.8)	3 (0.3)	0.443
Beta-lactams	89 (63.6)	847 (72.3)	**0.030**
Quinolones	25 (17.9)	208 (17.8)	0.978
Macrolides	34 (24.3)	421 (40.2)	**<0.001**
Oral anticoagulants ^g^	14 10.1)	86 (7.4)	0.255
Low-molecular-weight heparin	18 (12.9)	133 (11.3)	0.765
High-flow nasal cannula oxygen	22 (15.7)	54 (4.6)	**<0.001**
Non-invasive mechanical ventilation	7 (5.0)	51 (4.4)	0.724

CCI: Charlson Comorbidity Index; IQR: interquartile range; n (%): number of cases (percentage); qSOFA: quick sequential organ failure assessment. ^a^ Non-atherosclerotic heart disease comprises atrial fibrillation and/or heart failure. ^b^ Atherosclerotic cardiovascular disease comprises cerebrovascular, coronary, and/or peripheral vascular disease. ^c^ Chronic pulmonary disease comprises asthma and/or chronic obstructive pulmonary diseases. ^d^ Malignancy comprises solid tumors or hematological neoplasms. ^e^ Kidney disease is defined as an estimated glomerular filtration rate (eGFR) < 45 mL/min/1.73 m^2^ pursuant to the CKD-EPI equation. ^f^ Obesity is defined as a body mass index > 30 kg/m^2^. ^g^ Oral anticoagulant therapy (dicoumarin or direct oral anticoagulant). Statistically significant differences are indicated in bold.

**Table 2 jcm-11-03769-t002:** Clinical outcomes in very old patients (≥80 years) hospitalized with COVID-19 treated or not treated with remdesivir.

	Remdesivir (n = 140)	No Remdesivir (n = 1172)	OR (95% CI)	*p* Value
**Outcomes, n (%)**				
0-day all-cause hospital mortality	28 (20.0)	418 (35.7)	0.45 (0.29–0.69)	**<0.001**
Intensive care unit admission	5 (3.6)	15 (1.3)	2.85 (1.02–7.98)	**0.036**
Invasive mechanic ventilation	2 (1.4)	7 (0.6)	1.31 (0.96–1.12)	0.260
Readmission	14 (10.0)	90 (7.7)	1.33 (0.95–1.11)	0.338
Days of hospitalization, median (IQR) (non-survivors)	13.5 (8–24)	9 (6–14)	1.03 (1.02–1.04)	**<0.001**
Length of stay (days), median (IQR) (survivors)	15.5 (9–26)	9 (6–14)	1.04 (1.02–1.05)	**<0.001**

IQR: interquartile range; OR: odds ratio; CI: confidence interval. Statistically significant differences are indicated in bold.

**Table 3 jcm-11-03769-t003:** Multivariate logistic regression model for in-hospital mortality in very old (≥80 years) patients hospitalized with COVID-19 treated with remdesivir.

Independent Variables	Adjusted OR (95% CI)	*p* Value
Treatment with remdesivir	0.40 (0.24–0.66)	<0.001
Sociodemographic variables		
Age	1.02 (0.99–1.00)	0.176
Sex, male	1.31 (1.00–1.79)	0.047
Acquisition		
Community	1	
Nosocomial	2.42 (1.43–4.09)	0.002
Nursing Home	1.46 (1.11–1.91)	0.006
Degree of dependence		
Independent or mild	1	
Moderate	2.01 (1.44–2.81)	<0.001
Severe	2.46 (1.66–3.67)	<0.001
Comorbidities		
Baseline CCI	1.19 (1.11–1.27)	<0.001
Dementia	0.77 (0.54–1.08)	0.777
Symptoms and physical examination		
Duration of symptoms in days	0.99 (0.98—1.00)	0.154
qSOFA index ≥2	3.39 (2.26–4.87)	<0.001
Chest X-ray findings		
Normal	1	
Unilateral infiltrates	1.03 (0.68–1.56)	0.887
Bilateral infiltrates	1.73 (1.21–2.48)	0.002
Laboratory findings		
D-dimer	1.00 (1.00–1.00)	0.889
Other treatment		
Systemic corticosteroids	1.65 (1.13–2.41)	0.009
Tocilizumab	1.42 (0.79–2.55)	0.234
Beta-lactams	1.43 (1.05–2.01)	0.022
Macrolides	0.91 (0.68–1.22)	0.551
High-flow nasal cannula oxygen	6.84 (3.79–12.34)	<0.001

CCI: Charlson Comorbidity Index; OR: odds ratio; CI: confidence interval; qSOFA: quick sequential organ failure assessment.

## Data Availability

J.-M.R.-R. and R.G.-H. have full access to the data and are the guarantors for the data.

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
