# Peer review of "Remdesivir in Very Old Patients (≥80 Years) Hospitalized with COVID-19: Real World Data from the SEMI-COVID-19 Registry"

_jcm, 2022, doi:10.3390/jcm11133769_

Round 1

Reviewer 1 Report

The authors have evaluated the role of remdesivir in very old people with COVID 19. My comments are

1. Good data from secondary data evaluated the very old people also with geriatric evaluation.

2. In table 1 baseline characteristic, the comorbidities were differences between 2 group. Should be discussed in the discussion dementia, parkinsonisme, and others were severity factors of covid `19.  

  • DOI: 10.1016/j.parkreldis.2021.04.019  ; 
      • DOI: 10.1007/s00406-020-01205-z 

      3. The practical findings should be added in the discussion for clinical practice 

Author Response

Dear Reviewer,

Please find attached a revised version of our manuscript (ref. PNTD-D-22-00119), entitled “Remdesivir in very old patients (≥ 80 years) hospitalized with COVID-19. Real world data from the SEMI-COVID-19 Registry”, which we submit for consideration to publish in Journal of Clinical Medicine.

We have modified the original submission taking into consideration all suggestions All revisions are marked with track changes the changes are detailed below.

The authors have evaluated the role of remdesivir in very old people with COVID 19.

My comments are

1.Good data from secondary data evaluated the very old people also with geriatric evaluation.

Reply: Thank you for your comment.

2.In table 1 baseline characteristic, the comorbidities were differences between 2 group. Should be discussed in the discussion dementia, parkinsonisme, and others were severity factors of covid `19.  

DOI: 10.1016/j.parkreldis.2021.04.019  ;  

DOI: 10.1007/s00406-020-01205-z 

Reply: We appreciate the reviewer's comment. Unfortunately, we do not have data on Parkinson’s disease or Parkinsonism. However, we have included the following paragraph in the discussion section:

Certain risk factors, such as dependence and the presence of dementia, are associated with higher mortality in very old patients. In our study, few patients with moderate or severe dependence and dementia received remdesivir and they were likely a group of people who should have received the drug. Parkinson's disease and Parkinsonism are risk factors for worse outcomes in patients with COVID-19, as has been shown in the meta-analysis by Putri C., and these patients would probably have benefited from treatment. However, data on Parkinson’s disease and Parkinsonism were not collected in this study.

Also, we have included the following two references:

Hariyanto TI, Putri C, Situmeang RFV, Kurniawan A. Dementia is a predictor for mortality outcome from coronavirus disease 2019 (COVID-19) infection. Eur Arch Psychiatry Clin Neurosci. 2021 Mar;271(2):393-395. doi: 10.1007/s00406-020-01205-z.

Putri C, Hariyanto TI, Hananto JE, Christian K, Situmeang RFV, Kurniawan A. Parkinson's disease may worsen outcomes from coronavirus disease 2019 (COVID-19) pneumonia in hospitalized patients: A systematic review, meta-analysis, and meta-regression. Parkinsonism Relat Disord. 2021 Jun;87:155-161. doi: 10.1016/j.parkreldis.2021.04.019.

3.The practical findings should be added in the discussion for clinical practice.

We appreciate the reviewer's comment and have included the reviewer's suggestion in the study results.

It is unclear whether the good outcomes observed in elderly patients treated with remdesivir could be due to immunosenescence in this population [25]. If so, the earlier use of antiviral therapy in these patients could be another explanation of the better response to antivirals, in light of the fact that they accessed medical care earlier [26], when the efficacy of treatment is higher.

This work shows that remdesivir treatment in elderly patients was associated with lower mortality and that antiviral treatment in the first few days of illness reduced mortality in unvaccinated patients. Therefore, it can be expected that in vaccinated patients, which are the majority of those currently admitted to the hospital for COVID-19, treatment with remdesivir may improve the course of the disease in elderly patients.

We have also included the following two references:

  1. Perrotta, F.; Corbi, G.; Mazzeo, G.; Boccia, M.; Aronne, L.; D’Agnano, V.; Komici, K.; Mazzarella, G.; Parrella, R.; Bianco, A. COVID-19 and the elderly: insights into pathogenesis and clinical decision-making. Aging Clin Exp Res 2020, doi:10.1007/s40520-020-01631-y.
  2. Ramos-Rincón, J.M.; Bernabeu-Wittel, M.; Fiteni-Mera, I.; López-Sampalo, A.; López-Ríos, C.; García-Andreu, M.D.M.; Mancebo-Sevilla, J.J.; Jimenez-Juan, C.; Matía-Sanz, M.; López-Quirantes, P.; et al. Clinical Features and Risk Factors for Mortality Among Long-term Care Facility Residents Hospitalized Due to COVID-19 in Spain. J Gerontol A Biol Sci Med Sci 2022, 77, E138-E147, doi:10.1093/GERONA/GLAB305

We would like to thank you  for their time and overall positive comments. Most of the suggestions and comments were very valuable and we believe their constructive comments have significantly improved our submission. We hope the manuscript will now be suitable for publication in Journal of Clinical Medicine

Sincerely yours,

Jose Manuel Ramos MD, PhD

Reviewer 2 Report

The authors present interesting research findings. Although the data seems outdated due to several available reports on remdesvir and its discontinuity in treatment, the research methodology is flawless and certainly deserves publication.

Author Response

Reviewer 2

The authors present interesting research findings. Although the data seems outdated due to several available reports on remdesvir and its discontinuity in treatment, the research methodology is flawless and certainly deserves publication

Reply: We welcome your comments. We understand that these results are a bit outdated, but we believe that the experience during the covid-19 pandemic in unvaccinated patients may be useful to the readers of the journal.

Reviewer 3 Report

The authors retrospectively report an experience on treatment of COVID-19 in patients >80 years old with respiratory failure, with special attention to remdesivir, which was foud to be associated with lower mortality. The text is linear and cleary. Study design is well described. In the discussion it should be better speculated on why this observation was mad in elderly patients (immunosenescence and higher role of antiviral therapy? earlier access to medical care and better response to antiviral?), also considering the association of steroid with mortality which emerged from the multivariable analysis.

Author Response

Reviewer 3

The authors retrospectively report an experience on treatment of COVID-19 in patients >80 years old with respiratory failure, with special attention to remdesivir, which was foud to be associated with lower mortality. The text is linear and cleary. Study design is well described. 

Reply: Thank for your comment.

In the discussion it should be better speculated on why this observation was mad in elderly patients (immunosenescence and higher role of antiviral therapy? earlier access to medical care and better response to antiviral?), also considering the association of steroid with mortality which emerged from the multivariable analysis. 

Reply: Thank you for your comment. In this regard, we have added a sentence to the discussion on why elderly patients may have responded well to remdesivir treatment. 

It is unclear whether the good outcomes observed in elderly patients treated with remdesivir could be due to immunosenescence in this population [25]. If so, the earlier use of antiviral therapy in these patients could be another explanation of the better response to antivirals, in light of the fact that they accessed medical care earlier [26], when the efficacy of treatment is higher.

This work shows that remdesivir treatment in elderly patients was associated with lower mortality and that antiviral treatment in the first few days of illness reduced mortality in unvaccinated patients. Therefore, it can be expected that in vaccinated patients, which are the majority of those currently admitted to the hospital for COVID-19, treatment with remdesivir may improve the course of the disease in elderly patients.

We have also included the following two references:

25. Perrotta, F.; Corbi, G.; Mazzeo, G.; Boccia, M.; Aronne, L.; D’Agnano, V.; Komici, K.; Mazzarella, G.; Parrella, R.; Bianco, A. COVID-19 and the elderly: insights into pathogenesis and clinical decision-making. Aging Clin Exp Res 2020, doi:10.1007/s40520-020-01631-y.
26.     Ramos-Rincón, J.M.; Bernabeu-Wittel, M.; Fiteni-Mera, I.; López-Sampalo, A.; López-Ríos, C.; García-Andreu, M.D.M.; Mancebo-Sevilla, J.J.; Jimenez-Juan, C.; Matía-Sanz, M.; López-Quirantes, P.; et al. Clinical Features and Risk Factors for Mortality Among Long-term Care Facility Residents Hospitalized Due to COVID-19 in Spain. J Gerontol A Biol Sci Med Sci 2022, 77, E138-E147, doi:10.1093/GERONA/GLAB305